# Localized hypoxia within the subgranular zone determines the early survival of newborn hippocampal granule cells

Christina Chatzi[1], Eric Schnell[2,3], Gary L Westbrook[1]*

[1]The Vollum Institute, Oregon Health and Science University, Portland, United States; [2]Department of Anesthesiology and Perioperative Medicine, Oregon Health and Science University, Portland, United States; [3]United States Department of Veterans Affairs, VA Portland Health Care System, Portland, United States

**Abstract** The majority of adult hippocampal newborn cells die during early differentiation from intermediate progenitors (IPCs) to immature neurons. Neural stem cells in vivo are located in a relative hypoxic environment, and hypoxia enhances their survival, proliferation and stemness in vitro. Thus, we hypothesized that migration of IPCs away from hypoxic zones within the SGZ might result in oxidative damage, thus triggering cell death. Hypoxic niches were observed along the SGZ, composed of adult NSCs and early IPCs, and oxidative byproducts were present in adjacent late IPCs and neuroblasts. Stabilizing hypoxia inducible factor-1$\alpha$ with dimethyloxallyl glycine increased early survival, but not proliferation or differentiation, in neurospheres in vitro and in newly born SGZ cells in vivo. Rescue experiments in $Bax^{fl/fl}$ mutants supported these results. We propose that localized hypoxia within the SGZ contributes to the neurogenic microenvironment and determines the early, activity-independent survival of adult hippocampal newborn cells.

*For correspondence: westbroo@ohsu.edu

Competing interests: The authors declare that no competing interests exist.

## Introduction

In the subgranular zone (SGZ) of the adult hippocampal dentate gyrus (DG), one of the two neurogenic niches in the adult mammalian brain, new neurons are continuously generated throughout adulthood (*Taupin and Gage, 2002*; *van Praag et al., 2002*). Adult hippocampal neurogenesis is a highly dynamic and regulated process that evolves slowly over several weeks (*Zhao et al., 2008*). Adult neural stem/ radial glia-like cells (Type I cells) reside at the interface between the SGZ and the hilus, and give rise to rapidly dividing intermediate progenitors (Type 2 cells). These cells migrate a short distance into the SGZ and progressively (via Type 2a and Type 2b intermediate progenitors) differentiate into neuroblasts (Type 3 cells) (*Zhao et al., 2008*; *Fuentealba et al., 2012*; *Bonaguidi et al., 2012*). Neuroblasts then exit the cell cycle and become immature granule cell neurons, which in turn migrate into the granule cell layer and incorporate into the pre-existing functional hippocampal circuits (*Toni et al., 2008*).

In the young adult mouse DG as many as 4000 new cells are born daily, but only a subset (~30%) survive at 4 weeks post-mitosis to become mature granule cell neurons (*Dayer et al., 2003*; *Kempermann, 2003*; *Kempermann et al., 2006*; *Sierra et al., 2010*). The survival of adult-generated granule cells exhibits two critical periods; an early one during the transition from transient amplifying progenitors to neuroblasts and a later one during the integration of the immature neurons (*Dayer et al., 2003*; *Kempermann, 2003*; *Sierra et al., 2010*; *Mandyam et al., 2007*). The early phase is associated with phagocytosis of apoptotic cells by microglia (*Sierra et al., 2010*). GABA-mediated-depolarization and NMDA-receptor-mediated neuronal activity regulate the later phase of survival, at two and three weeks after mitosis respectively (*Jagasia et al., 2009*; *Tashiro et al.,*

**eLife digest** The hippocampus is a region in the mammalian brain that has been implicated in the formation of new memories. This process involves the birth of new neurons, which are created at a rate of ~4000 a day in the hippocampus of a young adult mouse. Yet only a fraction of these cells survive to form mature neurons. These cells die in two main waves – the first occurs days after they form, and the second several weeks later when as immature neurons they integrate into the brain. During this later wave, new neurons become active and survive if they connect with other nerve cells and die if they don't. But little is known about what causes the earlier wave of cell death.

The tissues that contain the precursors of new neurons often have lower oxygen levels compared to other tissues. This means that when these cells start to become neurons and leave these sites, they have to face higher levels of oxygen and may undergo "oxidative" damage. This led Chatzi et al. to ask whether such oxidative damage might cause the early loss of new neurons in the hippocampus.

First, the part of the hippocampus that contains the precursor cells (called the subgranular zone or SGZ) was found to have patchy areas of low oxygen. Further experiments then revealed that chemicals that may cause oxidative damage were present in the nearby cells that had already started on the path to become new neurons.

Chatzi et al. then tested whether chemically stabilizing a protein called Hypoxia Inducible Factor-1α (or HIF1α for short), which naturally helps cells to adapt to low oxygen environments, might increase the survival of the cells in the SGZ. Higher levels of HIF1α did indeed increase the survival of these cells.

These findings suggest that newborn cells in the SGZ walk a tightrope between a low oxygen environment that supports the early precursors and the surrounding higher oxygen levels that can be toxic to those cells that start to become neurons. Further studies of the proteins and molecules that act downstream of HIF1α could shed light on ways to enhance the survival of these newly-generated neurons.

2006). However, less is known about the mechanisms responsible for the early death of newborn granule cells which occurs during the first days post mitosis, as cells exit the specialized microenvironment (NSC niche) of the adult SGZ.

There is increasing evidence that differential oxygen tensions may be an important component of the NSC niche and that oxygen sensing can regulate the proliferation, survival and differentiation of neural stem cells and progenitors (*Panchision, 2009*; *de Filippis and Delia, 2011*; *Mazumdar et al., 2010*). Thus, we investigated hypoxia within the SGZ, and whether stabilizing Hypoxia Inducible Factor-1 α (HIF-1α), the master regulator of oxygen homeostasis, influences the early survival of the adult hippocampal newborn cells.

## Results

### Detection of hypoxic niches and oxidative damage in the adult SGZ

We assessed for hypoxia within the SGZ of the adult DG following intraperitoneal injection of the hypoxia marker pimonidazole hydrochloride (Hypoxyprobe, *Figure 1B*) (*Varia et al., 1998*). Pimonidazole (PH) is reductively activated and binds only to cells that have oxygen concentrations less than 14 μM, equivalent to a $pO_2$ of 10 mm Hg (1.3% $O_2$) (*Raleigh et al., 1998*). Pimonidazole labeled hypoxic cells or groups of cells along the inner border of the SGZ (*Figure 1B*). Double labeling with selective markers revealed that the majority (71.7 ± 10.4%) of the hypoxic cells were radial glia (GFAP+) and a smaller percentage (28.3 ± 9.5%) were early intermediate progenitor cells (Tbr2+/Dcx-) (*Figure 1C,D*) in close proximity to the SGZ. In contrast, none of the hypoxic cells (0%) expressed the neuroblast marker doublecortin (DCX, *Figure 1C,D*).

Shortly after their generation, proliferating intermediate precursors translocate away from the proximal domain and into the intermediate domain (*Fuentealba et al., 2012*). Thus, we hypothesized that migration away from hypoxic zones might result in oxidative damage as diagrammed in

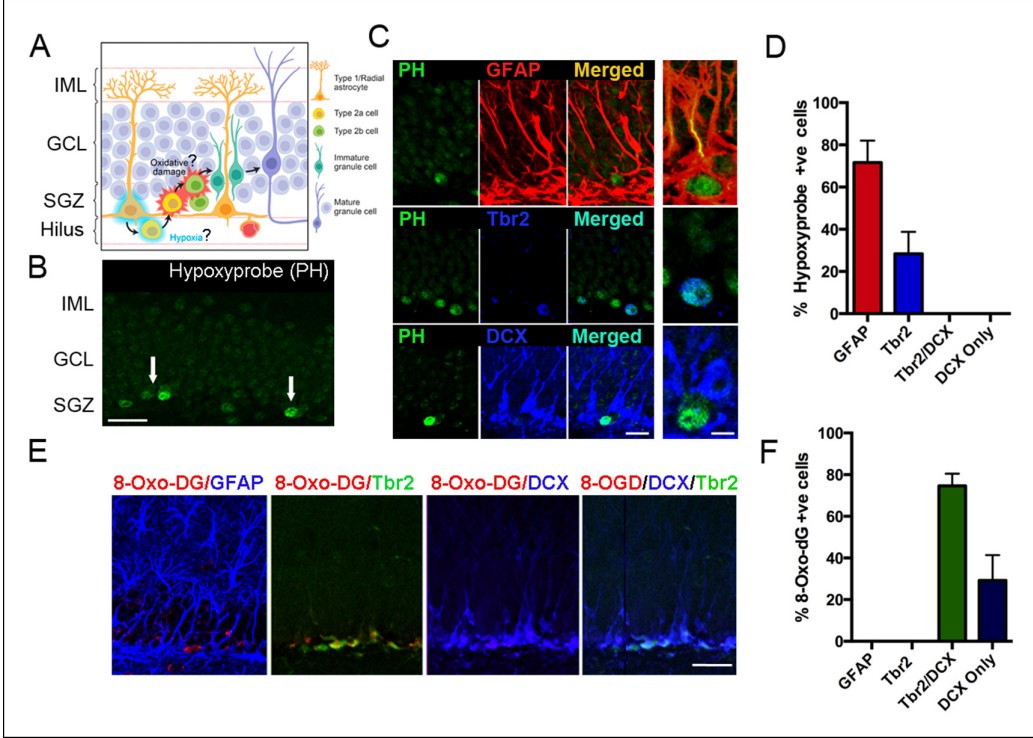

**Figure 1.** Detection of hypoxia and oxidative stress in SGZ niches of the adult DG. (**A**) Schematic diagram of experimental design. (**B**) Immunostaining with pimonidazole hydrochloride marks hypoxic areas (white arrows) within the adult SGZ (scale bar: 20 µm). (**C**) Pimonidazole-positive cells colocalized with stem cell marker GFAP (red, top row), intermediate progenitor marker Tbr2 (blue, middle row) but not with neuroblast marker DCX (blue, bottom row). The enlargement in the rightmost panel in the top row indicates a hypoxic neural stem cell (green) that colocalizes with a GFAP+ radial glial cell (yellow). A smaller fraction of early progenitors (Tbr2+) also were pimonidazole-positive (middle row, rightmost panel). Scale bar: 15µm left panel, 5 µm right panel. (**D**) Quantification of SGZ cells types expressing pimonidazole. (**E**) Oxidized 8-deoxyguanosine (8OHdG), a marker of oxidized nucleic acids, was not detected in GFAP positive neural stem cells, but in Tbr2- and DCX-expressing intermediate progenitors and neuroblasts (scale bar: 25 µm). (**F**) Quantification of SGZ cell types expressing oxidized 8-deoxyguanosine. For all quantifications data are plotted as mean ± SD.

*Figure 1A*. Oxidative byproducts were highly localized in cells within the SGZ as assayed by 8-oxo-7,8-dihydro-2'-deoxyguanosine (8-OHdG) labeling, an established biomarker of nucleic acid oxidative damage (*Figure 1E*). Interestingly, phenotypic analysis revealed that 74.6 ± 5.8% of the 8-OHdG positive cells were late intermediate progenitors (Tbr2+/DCX+) and 29.2 ± 12.2% were neuroblasts (DCX+ only) (*Figure 1E,F*). Thus, a subset of newborn cells in the SGZ undergoes oxidative damage during their early migration and differentiation.

## Stabilizing hypoxia inducible factor-1α

The most critical survival period for adult-generated dentate granule cells occurs during the first few days post mitosis, as they differentiate from late intermediate progenitors to neuroblasts (*Dayer et al., 2003*; *Kempermann, 2003*; *Sierra et al., 2010*; *Mandyam et al., 2007*). In order to test the role of oxidative damage in this early phase of apoptosis, we used the hypoxia mimetic agent, Dimethyloxallyl glycine (DMOG) (*Harten et al., 2010*). Hypoxia Inducible Factor-1 α (HIF-1α), the master regulator of oxygen homeostasis, is enzymatically degraded under normoxia by propyl hydroxylases (*Semenza, 2001*). DMOG suppresses propyl hydroxylase, thus stabilizing/enhancing HIF-1α activity under normoxic conditions (*Harten et al., 2010*).

We treated adult mice with DMOG (50 mg/kg once daily) or vehicle for 3 days. Consistent with stabilization of HIF-1α by DMOG, mRNA levels of HIF-1α as well as its downstream targets VEGF, EPO and LEF-1, were significantly higher after 3 days of DMOG treatment (*Figure 2A*). As expected

given the effect on downstream targets, DMOG treatment also increased HIF-1$\alpha$ protein levels (*Figure 2B*, p = 0.03). The increase in *HIF1$\alpha$* mRNA levels occurs because of autoregulation of transcription by hypoxia response elements in the *HIF1$\alpha$* promoter (*Iyer et al., 1998*). Hypoxia also increases phosphorylation of Akt (Ser[473]), a serine/threonine kinase that promotes cell survival and reduces apoptosis (*Beitner-Johnson et al., 2001*). Double immunohistochemistry with an antibody that specifically recognizes phosphorylated Akt together with an antibody against the neuroblast/immature neuron marker doublecortin (DCX) (*Figure 2C* left panels), revealed that DMOG treatment induced a two-fold increase in the number of phospho-AKT+/DCX+ cells relative to vehicle treated animals (p = 0.01, n = 3 animals, *Figure 2C* right panel), also consistent with the hypoxia mimetic action of DMOG.

## DMOG increases survival, but not proliferation or differentiation

Hypoxia in vitro influences neural precursors' proliferation, differentiation and survival (*Panchision, 2009*; *de Filippis and Delia, 2011*). There was no significant effect of DMOG on the net proliferation of newborn cells at 3 dpi. (control group, 12901 ± 1870 Ki67+ cells/mm$^3$, n = 5 animals, DMOG group 11859 ± 2953 Ki67+ cells/mm$^3$, n = 5 animals, p = 0.5, *Figure 3A,B,E*). Similarly, there was not a detectable effect on the total density of Tbr2+ cells (control group: 5776 ± 681 cells/mm$^3$, DMOG: 7331 ± 1381 cells/mm$^3$, p = 0.07, n = 5 animals). At 3 dpi. 43.9 ± 5.8% of the proliferating cells were neural stem cells, nestin-only expressing cells, and 55.3 ± 7.1% were intermediate progenitors expressing both nestin and DCX (n = 4 animals). DMOG did not cause a shift in the proliferative populations of SGZ progenitor at 3 dpi. (41.9 ± 9.53% Ki67+/Nestin+, 58.1 ± 8.33 Ki67+/Nestin+/DCX+, 2-way ANOVA, n = 4 animals, p = 0.56, *Figure 3C,D,F*). The composition of the progenitor subtypes, measured as a percentage of BrdU+ cells was unaffected by DMOG treatment at 3, 7, 14, 21 and 28 days post-injection (2-way ANOVA, no interaction between vehicle and DMOG groups, p >0.99). This data is summarized in *Figure 3G*. Specifically, at 3 dpi. BrdU+ cells consisted mainly of late intermediate progenitors (Tbr2+/DCX+) and neuroblasts (Tbr2-/DCX+), and a small number of early intermediate progenitors (Tbr2+/DCX-). At 7 dpi., the proportion of BrdU+ cells colabeled with Tbr2+/DCX+ decreased, whereas the proportion expressing only DCX+ increased, reflecting a shift towards immature neurons. By 14 dpi. Tbr2+ cells were not detected in either group with the majority of the BrdU+ cells expressing DCX. In the subsequent two weeks there was a significant decrease in the number of BrdU+/DCX+ cells consistent with lineage progression to mature neurons. DMOG had no effect on the total volume of the dentate gyrus (Control: 0.61 mm$^3$ ± 0.02, n = 3 animals, DMOG: 0.59 ± 0.04, n = animals, p = 0.89), indicating that there were no macroscopic changes in the tissue.

To assay the effect of DMOG on early cell survival, dividing cells were pulse-labeled with BrdU in adult mice just prior to DMOG administration, and the number of BrdU-labeled cells in the dentate gyrus was counted at several timepoints post-BrdU injection (dpi) (*Figure 4A*). DMOG resulted in an increase of BrdU labeled nuclei in the SGZ (*Figure 4B*). Quantitative analysis of the BrdU+ cells showed a significant effect of DMOG on survival of newborn cells in the SGZ (2-way ANOVA, p = 0.0002, *Figure 4C*). One day of DMOG treatment did not alter the number of BrdU+ cells, consistent with no effect on proliferation (p = 0.49, n = 6 animals, *Figure 4C*). However, DMOG significantly increased the survival of BrdU+ cells by 3 days post-mitosis (p <0.0001, n = 12 animals). Interestingly, the relative increase in BrdU-labelled cells in DMOG persisted at later time points (7 dpi., p = 0.0017, n = 8 animals; 14 dpi., p = 0.003, n = 6; 21 dpi., p = 0.007, n = 6; 28 dpi., p = 0.01, n = 6), as assessed by the rate of BrdU+ cell loss between 3 and 28 dpi. (Comparison of fits, p = 0.5, F = 0.7). To delineate the effective time window for the action on adult newborn granule cell survival by 28 dpi., dividing cells were labeled with BrdU at day zero, then exposed to DMOG from 0–3, 0–7 or 7–14 days (*Figure 4D*). Exposure to DMOG for the first three day post-mitosis resulted in a 33% increase in survival (p = 0.023, n = 3), whereas exposure for 7 days did not yield a significant further increase (p = 0.23, n = 3). However, DMOG administration from day 7–14, a critical period during which the newborn cells begin to integrate into the hippocampal network, had no effect on their survival (p = 0.76, n = 3). The differentiation of BrdU+ cells into mature neurons (NeuN+) by 28 dpi. was not affected by exposure to DMOG for either the first 3 or 7 days post mitosis (BrdU+NeuN+/BrdU+: 0–3 days, Vehicle: 96 ± 4%, DMOG: 95 ± 2%, n = 3; 0–7 days, Vehicle: 98 ± 3%, DMOG: 96 ± 2%, n = 3, 2-way ANOVA p = 0.8). The above results indicate that the hypoxia mimetic agent DMOG does not alter the proliferation or differentiation of newborn cells in the adult DG, but rather

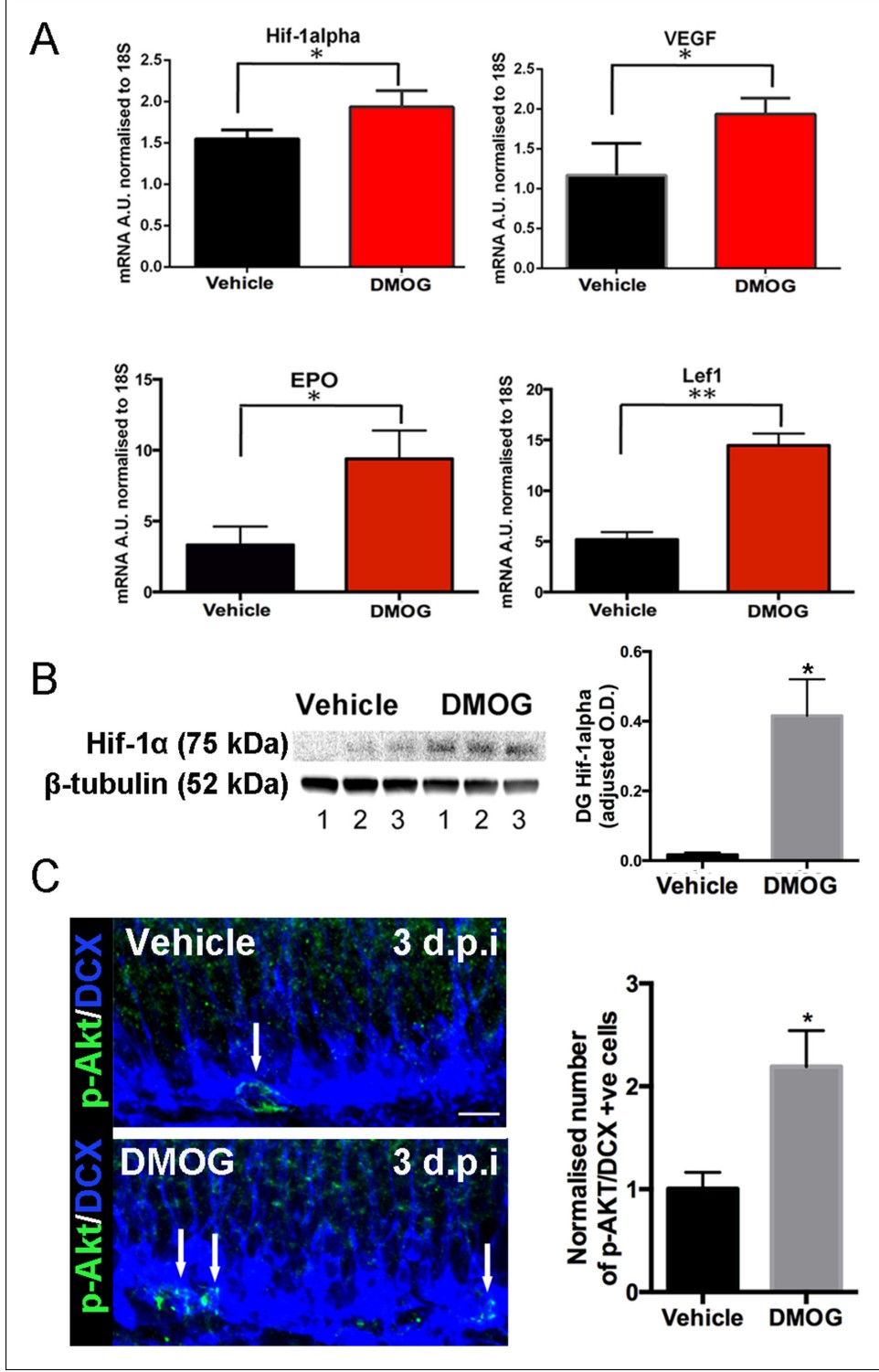

**Figure 2.** DMOG stabilized and activated Hif-1α signaling in vivo. (**A**) DMOG treatment elevated mRNA levels of HIF1-α and its downstream targets. HIF1-α, VEGF, EPO and Lef1 in the microdissected DG of animals treated with DMOG for 3 days. (HIF1-α, p = 0.04, n = 3; VEGF, p = 0.03, n = 3, EPO = 0.02, n = 3, Lef1, p = 0.002, n = 3). Data are mean ± SD. (**B**) Representative western blots of DG protein extracts probed with antibody against HIF1-α, from animals treated for 3 days with vehicle or DMOG (n = 3 each group, left panel). Semiquantitative densitometry for HIF1-α protein normalized to β-tubulin levels (right panel). HIF1-α DG protein levels were significantly elevated in DMOG treated animals. Data are mean ± SD, p = 0.03. (**C**) DMOG increases phosphorylation of Akt (Ser[473]) in DG
*Figure 2. continued on next page*

*Figure 2. Continued*

newborn cells. Representative images of adult DG sections stained with anti-phospho-Akt (green) and anti-DCX (blue) after 3 days treatment with vehicle or DMOG. Note the higher density of phospho-Akt positive cells in the SGZ of DMOG treated animals (below) compared to vehicle treated controls (above) (scale bar: 8 μm). Data are mean ± SD, p = 0.01.

specifically promotes their survival during the first few days post-mitosis, during their transition from intermediate progenitors to neuroblasts.

Apoptotic newborn cells in the DG are rapidly phagocytosed and cleared by unchallenged microglia (*Sierra et al., 2010*); rendering apoptosis hard to detect and evaluate with apoptotic markers such as TUNEL and activated-Caspase-3. Because the pro-apoptotic gene *Bax* mediates programmed cell death of adult-generated hippocampal cells (*Sun, 2004*; *Sahay et al., 2011*), we hypothesized that DMOG treatment should not affect the survival of cells lacking *Bax*. To test this

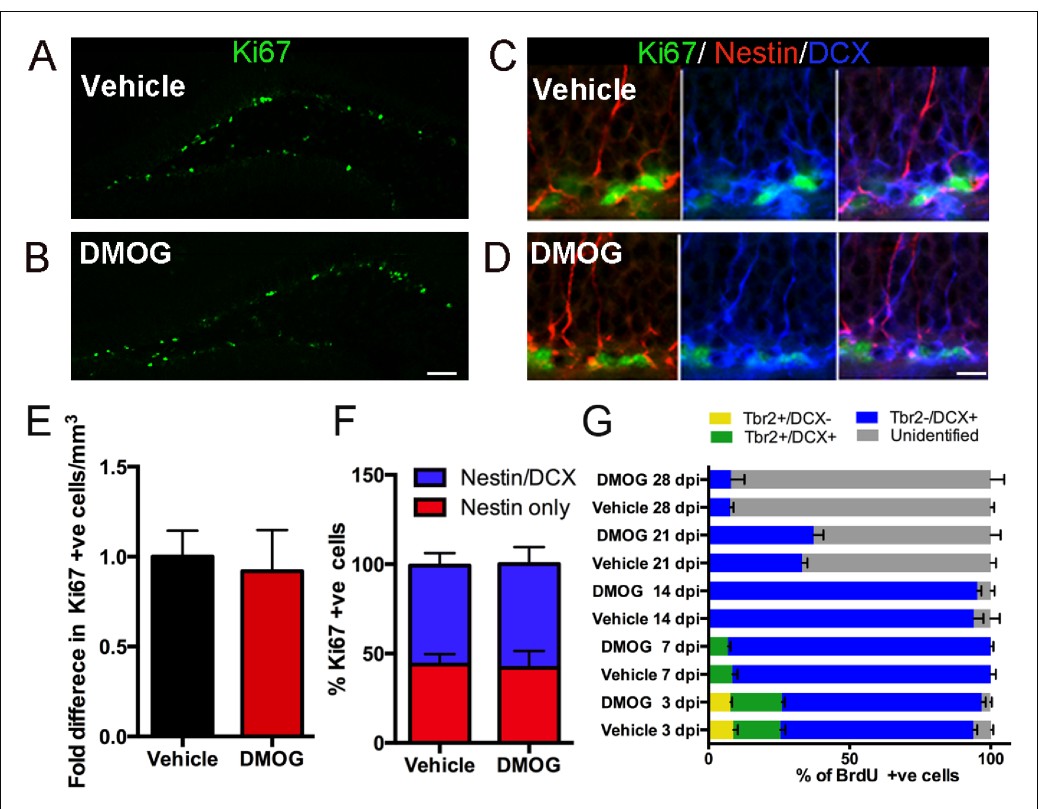

**Figure 3.** DMOG does not affect the proliferation and differentiation of 3 day old cells in the adult DG. (**A,B**) Representative images of proliferating cells Ki67+ cells in the SGZ (A, B, scale bar: 100 μm). (**C,D**) Triple labeling with Ki67/ Nestin/DCX (C, D, scale bar: 10 μm). (**E**) The density of proliferating cells (Ki67+) was comparable between vehicle controls and DMOG treated animals. (**F**) The proportion of the DG proliferative progenitors remained unaltered following DMOG administration. (**G**) To analyze the phenotype of Brdu+ cells, brains were collected 3, 7, 14, 21 and 28 days after two pulses of BrdU (300 mg/kg with a 4 hr interval between doses) and triple labeled with BrdU/Tbr2/DCX. DMOG treatment did not affect the composition of the SGZ progenitor subtypes at any of the examined time-points. For all quantifications data are plotted as mean ± SD. The percentages at each time point are as follows: 3 dpi.: Control: 8.8 ± 3.2% Tbr2+/DCX-, 16.78 ± 3.6% Tbr2+/DCX+, 68.4 ± 2.7 Tbr2-/DCX+; DMOG: 7.8 ± 1.1% Tbr2+/DCX-, 18.8 ± 1.9% Tbr2+/DCX+, 70.7 ± 2.9% Tbr2-/DCX+. 7 dpi.: Control: 8.4 ± 3.2% Tbr2+/DCX+, 91 ± 3.1% Tbr2-/DCX+; DMOG: 6.8 ± 1.8% Tbr2+/DCX+, 93.2 ± 1.7%. 14 dpi.: Control, 94 ± 5.6% Tbr2-/DCX+; DMOG: 95 ± 2.5% Tbr2-/DCX+. 21 and 28 dpi.: Control: 21 dpi. 33 ± 3.2% Tbr2-/DCX+, 28 dpi. 7.6 ± 2% Tbr2-/DCX+; DMOG: 21 dpi 37.2 ± 6.2% Tbr2-/DCX+, 28 dpi. 7.9 ± 8.4% Tbr2-/DCX+.

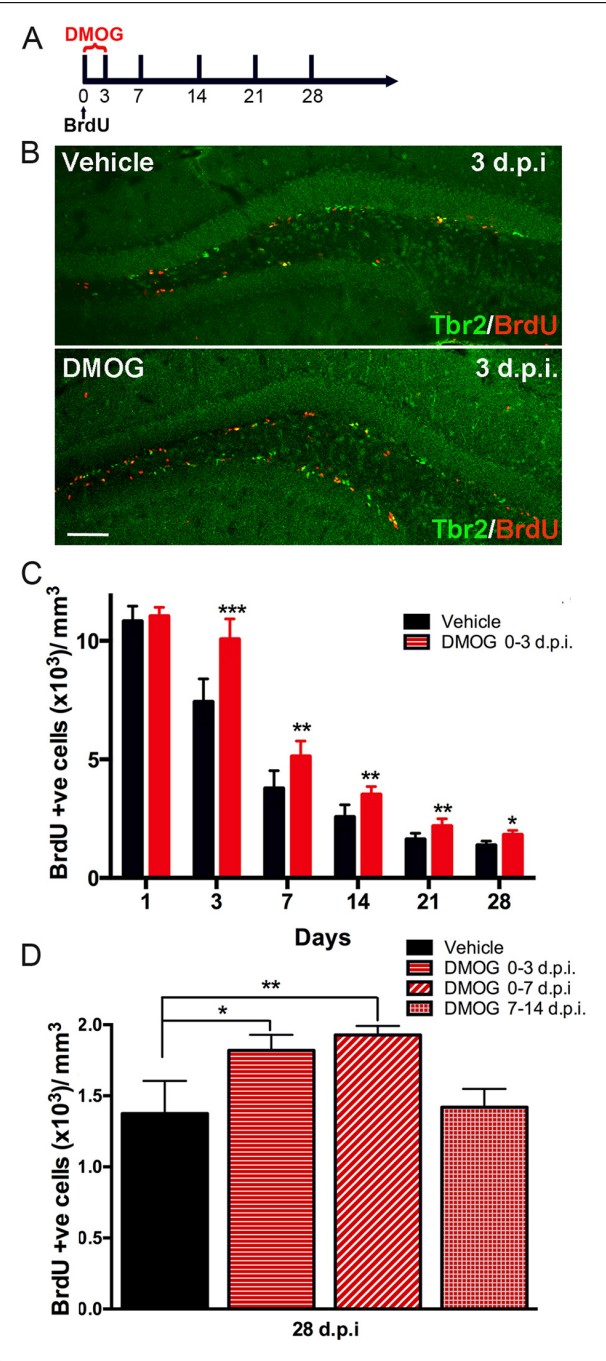

**Figure 4.** Hypoxia mimetic agent DMOG increases early survival of newborn cells in the adult SGZ. (**A**) The schema shows the experimental design for BrdU pulse labeling of newborn cells at day 0 followed by 3 days treatment with DMOG or vehicle and the different timepoints studied thereafter. (**B**) Representative immunofluorescence in sections of adult DG treated either with vehicle or with DMOG for 3 days double labeled with anti-BrdU and anti-Tbr2, which increased BrdU+ cells following DMOG treatment (scale bar: 100 μm). (**C**) Quantification of the BrdU positive cells along the time course of 28 days post injection (dpi) following 3-day treatment with vehicle or DMOG. (**D**) Quantification of the survival of BrdU positive cells at 28 dpi. following different DMOG treatment periods. For all quantifications data are plotted as mean ± SD (*p <0.05;**p <0.01;***p <0.001).

idea, the dentate gyrus of wildtype and *Bax<sup>fl/fl</sup>* mice was co-injected with a nuclear-Cre-GFP encoding retrovirus together with a control retrovirus encoding mCherry (*Figure 5A*). These MMLV-based retroviral constructs selectively target the same population of adult-born granule cells (*van Praag et al., 2002*), and could be used to count the numbers of labeled adult-born cells and selectively inhibit Bax expression from Cre-retrovirus infected cells. The ratio of nuclear-Cre-GFP expressing cells to the mCherry-expressing cells in *Bax<sup>fl/fl</sup>* animals as early as 7 dpi., provides a measure of survival of newborn cells, independent of possible variation in injection sites and viral titers. The ratio of Cre-GFP/ mCherry cells was increased in vehicle treated *Bax<sup>fl/fl</sup>* compared to WT animals at 7 dpi. (*Figure 5B,C*, 31.7 ± 10.3%, p = 0.002), indicating that loss of Bax augments the survival of adult-born cells at one week post-mitosis. Treatment with DMOG for 7 days post injection decreased the Cre-GFP/ mCherry ratio compared to *Bax<sup>fl/fl</sup>* vehicle treated animals (*Figure 5B,C*, 30 ± 9.4%, p = 0.03), consistent with a DMOG-mediated increase in survival in mCherry+ cells. Together our results strongly suggest activation of HIF-1α signaling enhances the early phase of survival of the newborn cells in the adult DG.

## Role of neuronal activity

Neuronal survival can be activity-dependent (*Jagasia et al., 2009*; *Tashiro et al., 2006*), thus we investigated whether newborn granule cells have received synaptic innervation by three days post-mitosis, the time point at which DMOG maximally increased survival. Acute slices from wildtype

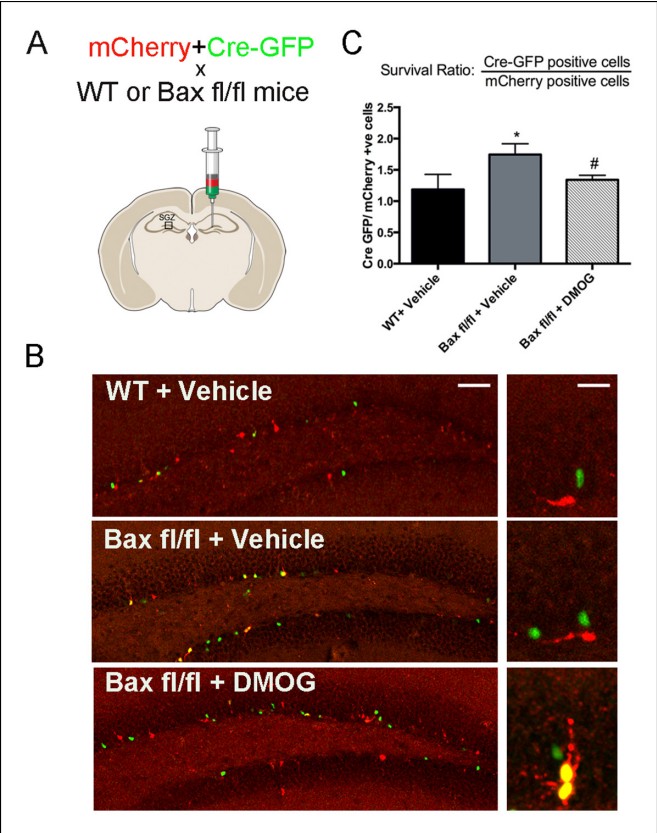

**Figure 5.** DMOG mimics the ablation of pro-apoptotic gene Bax in promoting survival of SGZ newborn cells. (**A**) A fixed ratio mixture of two retroviruses (encoding nuclear Cre-GFP and mCherry) was injected into the adult DG of WT and *Bax<sup>fl/fl</sup>* mice. Survival of the newborn cells was analyzed at 7 days post injection following treatment with either vehicle or DMOG. (**B**) Representative images of virus-labeled cells from co-injection experiments in WT mice treated with vehicle, *Bax<sup>fl/fl</sup>* mice treated with vehicle and *Bax<sup>fl/fl</sup>* mice treated with DMOG (scale bar: left 100 μm, right 20 μm). (**C**) The ratio of Cre-positive to mCherry-positive cells at 7 dpi was used as a measure cell survival (see text). Data are plotted as mean ± SD. *: WT+ Vehicle vs *Bax<sup>fl/fl</sup>* + Vehicle, p = 0.002; #: *Bax<sup>fl/fl</sup>* + Vehicle vs *Bax<sup>fl/fl</sup>* + DMOG, p = 0.03; WT+ Vehicle vs *Bax<sup>fl/fl</sup>* vs DMOG: not significant.

animals were prepared 3 days after retroviral infection and GFP-expressing 3 day old cells were identified using fluorescence microscopy and recorded using whole-cell voltage clamp techniques. Labeled cells were predominantly located in the subgranular zone as expected, with passive membrane properties consistent with immature neuronal precursors ($R_{input}$ = 13.2 ± 1.9 GΩ, $C_m$ = 4.8 ± 0.5 pF, n = 11 cells), including a fast inward current in response to a depolarizing voltage step consistent with a $Na^+$ spike in 9 of 11 cells (*Figure 6A*). Strikingly, none of these cells in wildtype animals had any spontaneous post-synaptic currents (PSCs) in >5 min of continuous recording per cell (n = 11), compared to an sPSC frequency of 0.55 ± 0.15 Hz in adjacent mature cells (n = 4). Furthermore, none of the 3 day-old cells had any evoked synaptic responses in the middle molecular layer ( >ten 10-second 8 Hz trains per cell, n = 11; *Figure 6A*), despite robust extracellular field potentials in these same slices at the same stimulation intensity (n = 7, data not shown) or large PSCs in neighboring mature cells (n = 4; *Figure 6B*). As expected for this feedforward circuit, the same middle molecular layer stimuli also produced repetitive spiking in hilar neurons (data now shown), indicating that hilar mossy cells or GABAergic interneurons were activated by our stimulation but were not functionally connected to 3 day-old cells. Together, these results indicate that there is minimal, if any, synaptic innervation at this maturational stage.

Likewise DMOG did not affect neuronal activity in immature or mature DG granule cells as measured by immunohistochemistry for the immediate early genes pCREB or c-Fos (*Figure 6C,D*), which are widely used as surrogate markers for neural activity (*Jagasia et al., 2009*; *Lonze and Ginty, 2002*). Thus the action of DMOG in our experiments cannot be attributed to a non-cell autonomous effect on neuronal activity of granule cells. Labeled cells were quantified as follows: pCREB - (control group, 122535 ± 19273 p-CREB+ cells/mm³, n = 4 animals, DMOG group 134051 ± 9733 p-CREB+ cells/mm³, n = 4 animals, p = 0.35); c-Fos - (control group, 20175 ± 1025 c-Fos+ cells/mm³, n = 4 animals; DMOG group 20646 ± 2766 c-fos+ cells/mm³, n = 4 animals; p = 0.9). DMOG treatment also did not affect neuronal activity in the hilus as assayed by double immunohistochemistry with c-Fos and the GABAergic neuronal marker glutamic acid decarboxylase-64 (Gad-67). We detected only sparse activation of hilar mossy cells (Fos+ only) or hilar interneurons (Fos+/GAD67+) under basal conditions, which was unaffected by DMOG (Vehicle, Fos+: 144 ± 55 cells/mm³, Fos+/GAD67 +: 14 ± 16 cells/mm³, n = 4; DMOG, Fos only+: 116 ± 14 cells/mm³, n = 4, Fos+/GAD67+: 13 ± 14 cells/mm³, n = 4, p = 0.4) These results indicate that stabilization and activation of HIF-1a signaling by DMOG rescues early survival of newborn cells in an activity-independent manner.

## Effect of hypoxia on adult DG-derived neurospheres

To further test the role of hypoxia, we used the neurosphere assay to examine survival and proliferation of adult DG progenitors (*Azari et al., 2010*; *Deleyrolle and Reynolds, 2009*; *Louis et al., 2013*). Adult DG neurospheres were generated and cultured for 7 days at ambient oxygen tension (21% $O_2$), in the presence of DMOG (100 μM) or in hypoxic conditions (2.5% $O_2$). Neurospheres under all three conditions exhibited normal morphology, each containing committed progenitors that co-expressed nestin and doublecortin (*Figure 7A*). Treatment with DMOG or reducing $O_2$ levels to 2.5% increased the number of neurospheres, a measure of cell survival (p = 0.01 and p = 0.002 respectively), but had no effect on the size of the neurospheres, a measure of cells' proliferation (p = 0.98, *Figure 7B,C*). Consistent with enhanced survival DMOG or hypoxia reduced the percentage of apoptotic cells in neurospheres as measured by caspase-3 immunoreactivity (normoxia: 28.1 ± 1.7%; DMOG: 17.7 ± 2.6%, p = 0.01; hypoxia: 11.4 ± 2.3%, p <0.0001, *Figure 7D–G*). The neurosphere assay also enabled us to test the sensitivity of the hypoxic marker pimonidazole hydrochloride (PH), which labeled groups of cells in the SGZ (*Figure 1B*). No pimonidazole binding was detected in neurospheres cultured in standard atmospheric culture conditions or in the presence of DMOG, whereas only a small percentage of PH+ cells (10.8 ± 4.4%), were detected in neurospheres cultured in 2.5% $O_2$ (*Figure 7H–K*), indicating that PH labeling occurs at $O_2$ ≤2.5%. Consistent with our in vivo experiments, our neurosphere data results strongly suggest that hypoxia is an essential factor for the survival of DG-derived intermediate progenitors in vitro.

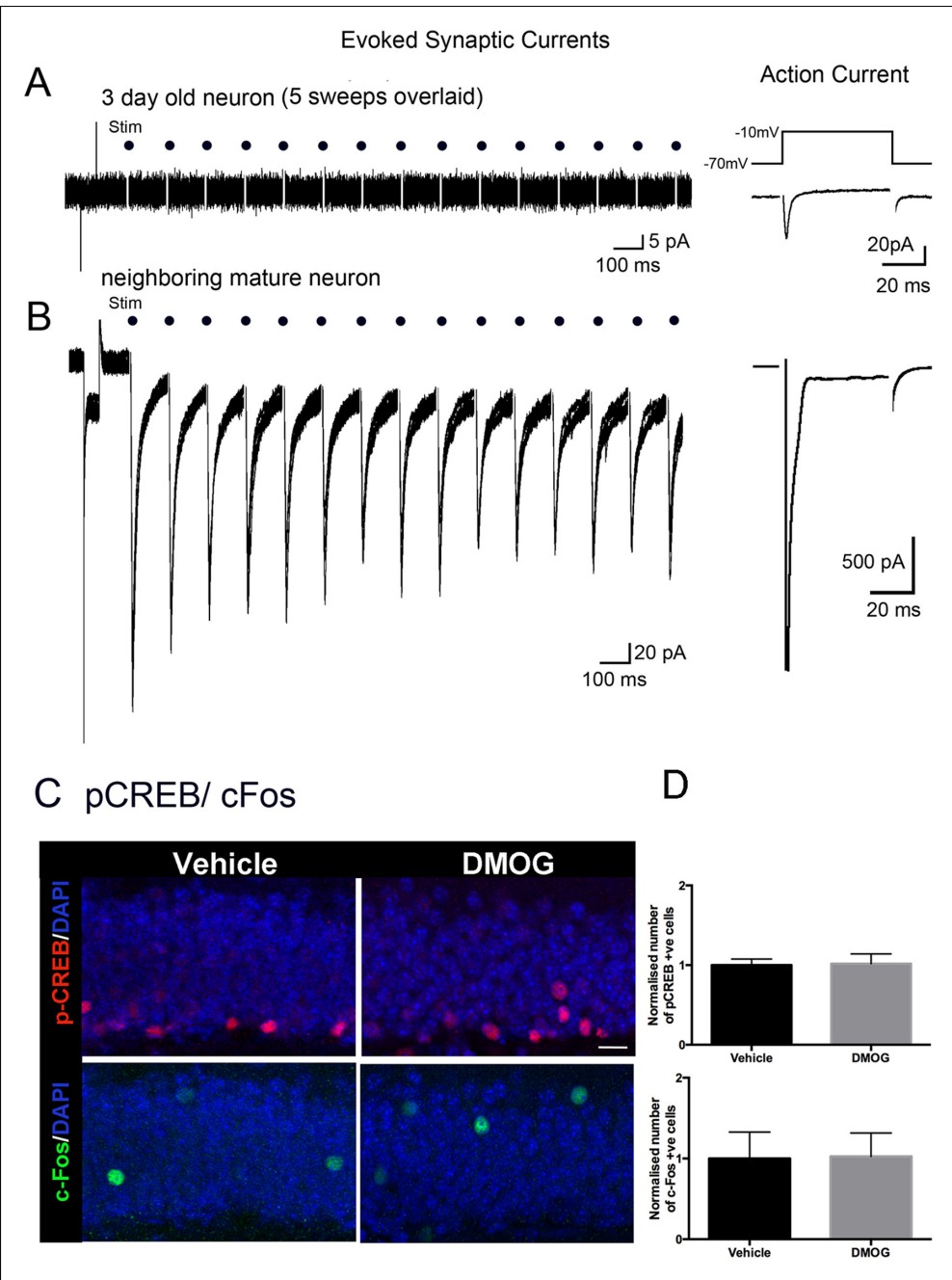

**Figure 6.** Adult-born granule cells lack synaptic responses 3 days post-mitosis. (**A**) At 3-days post-mitosis, granule cell lack responses to afferent stimulation. Five consecutive sweeps from a retrovirus-labeled 3-day-old cell, recorded in whole-cell voltage clamp mode while stimulating afferent inputs in the middle molecular layer. A -10 mV voltage step at the beginning of each sweep demonstrated the high input resistance of these cells, which lacked any postsynaptic responses to stimulation (dots). (**B**) A mature granule cell immediately adjacent to the cell in A had robust post-synaptic currents to the same stimulation. Right panels: Voltage-dependent sodium current evoked in a 3-day-old granule cell during a step from -70 to -10 mV, demonstrating the neuronal identity of the cell (right upper panel). The voltage-dependent sodium current recorded from a mature granule cell was much larger (right lower panel). (**C**) Representative confocal images of p-CREB+ and c-Fos+ cells in the DG of animals treated with vehicle or DMOG for 3 days (scale bar: 10 μm). (**D**) Quantification of the normalized number of p-CREB+ and c-Fos+ cells in the DG of DMOG treated animals relative to vehicle treated ones. DMOG treatment did not alter the number of p-CREB+ or c-Fos+ cells. Bar charts are mean ± SD.

## Discussion

### A hypoxic niche in the SGZ

The partial oxygen pressure ($pO_2$) and concentration in brain tissue is well below the 40 mm Hg in venous blood exiting the brain and is also heterogeneous (*Erecińska and Silver, 2001*). In rat hippocampus the $pO_2$ has been estimated to be 20.3 mmHg (*Erecińska and Silver, 2001*), whereas even more hypoxic regions have been detected in the SGZ of the adult mouse DG (*Mazumdar et al., 2010*). Using the hypoxia marker, pimonidazole hydrochloride, which labels cells with oxygen levels <10 mm Hg (1.3%), we found that SGZ stem-cell like-radial glia and early intermediate progenitors lie within hypoxic zones. Interestingly, not all cells in the SGZ were labeled, but were localized to narrow zones of a few cell diameters wide along the hilar border of the SGZ. Only a fraction of GFAP+ or Tbr2+ cells were pimonidazole-positive whereas the hypoxic cell did not label with the proliferating marker Ki67 (data not shown). Thus heterogeneous zones of tissue $pO_2$ may reflect some degree of cell-state specificity as well as local metabolic demand that changes constantly even within a small specific area/volume of the brain.

The two major determinants for tissue oxygen concentration are blood flow (oxygen supply) and oxygen consumption rate (cellularity). The latter may predominate in the SGZ with its high ratio of nuclei to blood vessels compared to other brain regions (*Mazumdar et al., 2010*). Although some neural stem cells are perivascular (*Ottone et al., 2014*; *Tavazoie et al., 2008*; *Palmer et al., 2000*), it remains to be determined whether these vessels carry highly oxygenated blood or how easily oxygen is distributed to the cell. Interestingly, a recent paper (*Sun et al., 2015*) revealed that within the SGZ zone only late amplifying progenitors (Tbr2+/DCX+) and horizontal early neuroblasts (Tbr2-/DCX+), among the entire population of SGZ progenitors are in direct contact with blood vessels.

Numerous adult stem cells reside in hypoxic niches, where they maintain a quiescent state depending predominantly on anaerobic glycolysis (*de Filippis and Delia, 2011*; *Mohyeldin et al.,*

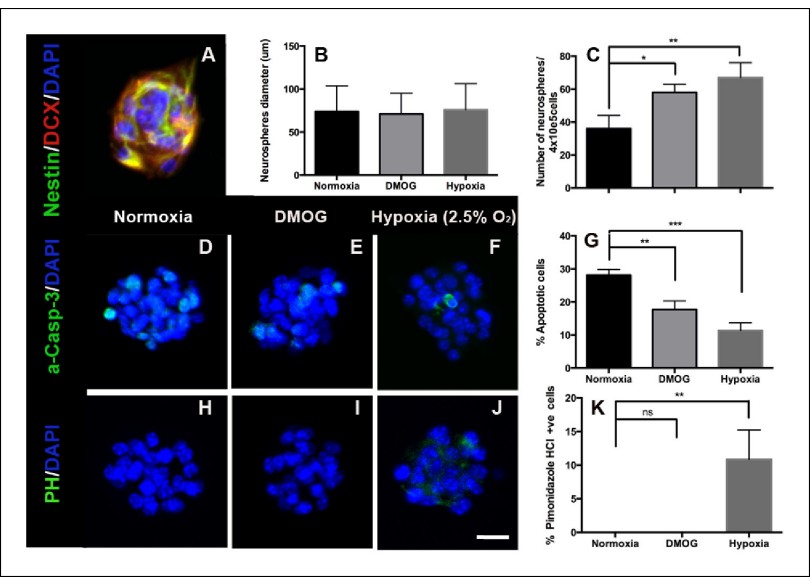

**Figure 7.** DMOG and Hypoxia increase survival of adult DG- derived neurospheres. (**A**) Representative immunofluorescence image of an adult DG-derived neurosphere, composed of nestin+ and DCX+ progenitors. Quantification of neurosphere diameter (**B**) and number (**C**) in cultures grown at normoxia, in the presence of DMOG or under hypoxia (2.5% $O_2$) for 7 days. (**D–F**) Representative immunocytochemistry for activated caspase-3 in neurospheres cultured under the 3 different experimental conditions. (**G**) Quantification of apoptotic (activated Caspase-3+) cells among the total number of cells in neurospheres exposed to normoxia, DMOG or hypoxia for 7 days. (**H–J**). Immunostaining with pimonidazole hydrochloride detected some hypoxic cells only in neurospheres cultured in 2.5% $O_2$ for 7 days. (Scale bar: 25 µm). (**K**) Quantification of pimonidazole hydrochloride-positive cells among the total cell number in neurospheres exposed to normoxia, DMOG or hypoxia. Bar charts are mean ± SD (*p <0.05; **p <0.01; ***p <0.001).

2010; *Platero-Luengo et al., 2014*). Stem cells exhibit different metabolic properties than their differentiated progeny that may promote 'stemness'. However, whether anaerobic metabolism is an adaptation to low oxygen levels in the specific niches in vivo or is an intrinsic stem cell property is still unclear. In either case, our results are consistent with two-photon phosphorescence direct in vivo measurements of local oxygen tension in the bone marrow, which revealed a hypoxic adult stem cell niche with high heterogeneity in local $pO_2$ (*Spencer et al., 2014*). Thus our data supports the existence of steep $pO_2$ gradients between the neurogenic niche and the granule cell layer.

### Tissue oxygen and apoptosis

Although a microenvironment with low oxygen concentration confers resistance to reactive oxygen species and cytotoxic stressors, stem cells and their progeny are susceptible to changes in redox status upon migration away from hypoxic zones (*Mohyeldin et al., 2010*; *Platero-Luengo et al., 2014*; *Suda et al., 2011*). Interestingly, we detected oxidative byproducts in intermediate progenitors and neuroblasts located adjacent to the hypoxic niches. This correlation suggests that migration of intermediate progenitors away from hypoxic zones leads to oxidative damage, and thus triggers an early phase of apoptosis. In vitro, hypoxic conditions (defined as <5% $O_2$) reduces apoptosis, promotes proliferation, and increases cultured embryonic, adult neural stem cells and neuronal progenitors (*Sierra et al., 2010*; *Shingo et al., 2001*; *Studer et al., 2000*; *Felling, 2006*; *Gustafsson et al., 2005*; *Clarke and van der Kooy, 2009*; *Chen et al., 2007*; *Bürgers et al., 2008*). Deletion of hypoxia-inducible factor-1 alpha subunit (HIF-1α) in vivo, a transcriptional activator mediating adaptive cellular responses to hypoxia, dramatically decreased adult hippocampal neurogenesis (*Mazumdar et al., 2010*). Additionally, conditional ablation of HIF1-α in adult mouse brain resulted in hydrocephalus, decreased neurogenesis (partly due to an increase in apoptosis) and deficits in spatial memory (*Tomita et al., 2003*). In our experiments, DMOG, an agent that stabilizes HIF1-α under normoxic conditions, reversed the early phase of cell death of newborn cells in the adult dentate gyrus. These results strongly support the idea that transition of progenitors from a hypoxic niche to normal tissue oxygen contributes to cell death.

### The time course of survival in the SGZ

Prior studies using mitotic markers such as BrdU found that only 30–50% of adult-born newborn neurons survive by 30 days post-mitosis (*Dayer et al., 2003*; *Kempermann, 2003*; *Mandyam et al., 2007*; *Jagasia et al., 2009*; *Tashiro et al., 2006*), but these studies often did not investigate survival during the first few days post-mitosis. Thus the prevailing idea, until recently, had been that adult-generated hippocampal granule cell survival is regulated mostly at the immature neuron stage (2–4 weeks post-mitosis), when is affected by neuronal activity and behavior (*Jagasia et al., 2009*; *Tashiro et al., 2006*; *Gage et al., 1999*). Our results support two critical periods for survival of newborn cells in the adult SGZ, early (1–4 days post-mitosis) and late (1–3 weeks post-mitosis) with most (2/3rds) of cell death occurring in the first week (*Dayer et al., 2003*; *Kempermann, 2003*; *Sierra et al., 2010*; *Mandyam et al., 2007*; *Jagasia et al., 2009*; *Tashiro et al., 2006*; *Mazumdar et al., 2010*). The early apoptosis is likely executed by members of the p53 and Bcl-2 protein families that are key players regulating the apoptotic machinery of adult DG neural precursors and newborn neurons (*Sun, 2004*; *Sahay et al., 2011*; *Cancino et al., 2013*; *Fatt et al., 2014*). Although programmed neuronal death is often activity-regulated, whether there is synaptic innervation during the first few days post-mitosis has been unclear (*Song et al., 2013*; *Esposito, 2005*). Our results show that at 3 days post-mitosis, adult-born cells demonstrate no detectable neural activity. Rather, we propose that the metabolic milieu of the adult SGZ, a previously underexplored variable of this highly specialized neurogenic niche, plays a critical role in the early survival of adult generated hippocampal granule cells.

## Materials and methods

### Animals

All procedures were performed according to the National Institutes of Health Guidelines for the Care and Use of Laboratory Animals and were in compliance with approved IACUC protocols at Oregon Health & Science University. Subjects were young adult (six weeks old) C57BL/6J (wildtype)

and B6;129-Bax[tm2Sjk] Bak1[tm1Thsn/J] (*Bax[fl/fl]*) transgenic mice (*Takeuchi et al., 2005*). Homozygotic *Bax[fl/fl]* mutants generated on a *Bak1* null background have exons 2–4 of *Bax* deleted following Cre-mediated recombination. The conditional deletion of *Bax* combined with the *Bak1* null allele greatly reduces apoptotic cell death and thus makes these mice useful in studies of apoptosis regulation (*Takeuchi et al., 2005*).

## Reagents

Hypoxic zones in the adult hippocampus were detected using pimonidazole hydrochloride kit (Hypoxyprobe) according to manufacturer's protocol. Pimonidazole hydrichloride in water was injected intraperitoneally at a dosage of 60 mg/kg. Animals were sacrificed at 1 hr and the brain removed following cardiac perfusion. Given that the half-life of pimonidazole is 22 min in mice, >99% of free drug had been cleared by the time of sacrifice, thus the pimonidazole imaged in our experiments was already bound at time of perfusion (*Walton et al., 1987*; *Williams et al., 1982*).

To examine survival of newborn cells in the adult hippocampus, mice were injected intraperitoneally with Bromodeoxyuridine (BrdU, Sigma-Aldrich, St. Louis, MO) at 300 mg/kg twice with a 4 hr interval between doses, and sacrificed at different time points. This pulse-chase protocol was chosen to saturate mitotic cell labeling within a single cell cycle as determined previously (*Cameron and Mckay, 2001*). To stabilize hypoxia-inducible factor 1-α, mice were treated with dimethyloxallyl glycine (DMOG, Cayman Chemicals, Ann Arbor, MI) at 50 mg/kg intraperitoneally daily for 3 or 7 days and animals were sacrificed at different time points. The comparison group received vehicle (30% DMSO). The primary antibodies used were: anti-pimonidazole hydrochrolide (1:100, Hypoxyprobe), anti-glial fibrillary acidic protein (GFAP; 1:1000, Dako, Denmark), anti-Tbr2 (Heffner lab), anti-double-cortin (DCX, 1:500, Millipore, Billerica, MA), Anti-8-Hydroxyguanosine (1:100, Calbiochem, Billerica, MA), anti-BrdU (1:500, Abcam), anti-phospho-Akt (1:100, Cell Signalling, Danvers, MA), anti-phospho-CREB (1:300, Santa Cruz), anti-c-fos (1:300, Santa Cruz, Dalla, TX), anti-NeuN (1:500, Sigma) and anti-GAD67 (1:500, Sigma).

## Immunohistochemistry and quantitation

Mice were terminally anesthetized according to IACUC-approved protocols, transcardially perfused with saline (4 ml, <30 sec) followed by 20ml of 4% paraformaldehyde (PFA), and brains were post-fixed overnight. Coronal sections (100μm thick) of the hippocampus were collected from each mouse and permeabilized in 0.4% Triton in PBS (PBST) for 30 min. Sections were then blocked for 30 min with 5% horse serum in PBST and incubated overnight (4°C) with primary antibody in 5% horse serum/PBST. Sections incubated with anti-BrdU were first incubated in 2N hydrochloric acid in potassium PBST for 30 min (37°C), washed twice and blocked with horse serum as described above. After extensive washing, sections were incubated with the appropriate secondary antibody conjugated with Alexa 488, 568 or 647 (Molecular Probes, Eugene, OR), for 2 hr at room temperature. They were then washed in PBST (2 × 10 min) and mounted with Dapi Fluoromount-G (SouthernBiotech, Birmingham, AL).

For quantification of immunopositive cells, six sections offering dorsal to ventral coverage of the dentate gyrus were stained from each animal. For the BrdU survival analysis, 6–12 animals were analyzed for each time point and condition, while for the rest of the immunofluorescence quantification 3–5 animals were analyzed per group. Slides were coded and imaged by an investigator blinded to experimental condition with a Zeiss LSM780 confocal microscope under a 10x 0.45NA, 20x or 40x 0.8NA lens. A 49 μm z-stack (consisting of 7 optical sections of 7μm thickness) was obtained from every slice. For evaluation of cells densities, positive cells within the dentate gyrus per z-stack were counted (Image J) and divided by the volume of the dentate gyrus. To determine the volume of the DG quantified the cross sectional area of the dentate gyrus (Image J) was multiplied by the section thickness. Cells densities from every stack were averaged per mouse and the results were pooled to generate mean values. For normalization the density of cells in individual DMOG treated animals was normalized to the mean cell density of the animals in the control group.

## Volumetric analysis of the DG

Measurements were taken in every sixth 49 μm coronal section stained with DAPI (Sigma-Aldrich). Sections were digitized at appropriate magnification and the areas of the dentate granule cell layer

(GCL) were measured using ImageJ software. Volumes (V) were calculated as $V = \Sigma A \cdot i \cdot d$, according to Cavalieri's principle, with A representing the sum of areas from both hemispheres of each section, $i$ the interval between the sections, and $d$ the section thickness, respectively.

## Quantitative real-time PCR

Total RNA was isolated from dentate gyri microdissected from animals treated with either vehicle or DMOG for 3 days. RNA samples were DNAse treated using the DNAfree kit (Ambion, Carlsbad, CA) followed by an additional ethanol precipitation. cDNAs were synthesized from 200 ng of RNA using random hexamer primers from the First Strand cDNA Synthesis Kit (Fermentas, Waltham, MA). The real time reaction was performed in triplicate using FastStart SYBR Green Master (Roche, Switzerland). Quantification was performed using the efficiency-corrected $\Delta\Delta$CT method. The following primers were used for qRT-PCR: HIF-1$\alpha$ sense CCATTCCTCATCCGTCAAATA anti-sense AAGTTCTTCCGGCTCATAAC, EPO sense CAGAGACCCTTCAGCTTCATATAG, anti-sense TCTGG-AGGCGACATCAATTC, VEGF sense ACACCCACCCACATACA, anti-sense TCCAGTGAAGACACC-AATAAC, Lef-1 sense AGAACACCCTGATGAAGGAAAG, antisense GTACGGGTCGCTGTTCATATT, 18S sense CGCGGTTCTATTTTGTTGGT, anti-sense TCGTCTTCGAAACTCCGACT.

## Western blot analyses

Microdissected dentate gyri from animals treated either with vehicle or DMOG for 3 days were mechanically homogenized in RIPA lysis buffer (Thermoscientific, Waltham, MA) and samples were spun for 10min at 4°C. The protein concentration in the supernatant was determined using the Bradford Protein Assay and supernatants were boiled at 95°C for 5min. Total protein samples were separated by SDS-PAGE and equal amounts of protein (30 μg) were loaded into a 10% gradient precast gel (Invitrogen). The separated proteins were transferred on a polyvinilydine difluoride membrane by electro transfer. The membrane was then incubated sequentially with the primary antibodies [mouse monoclonal HIF1$\alpha$ (Abcam, UK, Ab-1, 1:1000); mouse monoclonal β-tubulin (Developmental Studies Hybridoma Bank, Iowa City, IA, E7, 1:5000). Target proteins were detected using a chemiilluminescence ECL system (Genesys Technologies, Daly City, CA) after incubation with secondady antibody with conjugated horseradish peroxidase (anti-mouse IgG, HRP linked, Cell signalling). The blots were densitometrically analysed using Image J. The optical densities were calculated by normalising the ratio of target protein to β-tubulin.

## Retrovirus production and stereotaxic injections

Moloney Murine Leukemia Virus-based retroviral vectors require cell mitosis for transduction and were used to identify and manipulate adult-born granule cells. Retroviruses were created using a pSie-based viral backbone, and expression was driven by a ubiquitin promoter and followed by a woodchuck posttranscriptional regulatory element (*Tashiro et al., 2006*; *Luikart et al., 2011*; *Schnell et al., 2014*).

To study cell survival, groups of WT and *Bax*$^{fl/fl}$ mice (4–7 animals per condition) were injected with a 1:1 mixture of two viruses, the nuclear cre-GFP-expressing virus and the the mCherry-expressing virus, to control for viral injection coordinates and baseline cell survival. Mice were anesthetized using an isoflurane gas system (Veterinary Anesthesia Systems Co.) and placed in a Kopf stereotaxic frame fitted with a gas nose cone. A skin incision was made and holes were drilled at x: ± 1.1 mm, y: −1.9 mm from bregma. Using a 10 μl Hamilton syringe with a 30 ga needle and the Quintessential Stereotaxic Injector (Stoelting, Kiel, WI), 2 μl mixed viral stock (1 μl of each virus) was delivered at 0.25 μl/min at z-depths of 2.5 and 2.3 mm. The syringe was left in place for 1 min after each injection before being slowly withdrawn. The skin above the injection site was closed using veterinary glue. Animals received post-operative lidocaine and drinking water containing children's Tylenol. The number of GFP-expressing cells was divided by number of mCherry expressing cells for each animal.

## Electrophysiology

Live hippocampal slices were prepared from virus-injected animals and whole cell voltage-clamp recordings to asses synaptic activity were performed from retrovirus-labeled 3 day old granule cells using a cesium gluconate-based internal. Excitatory afferents and feed-forward inhibition were stimulated with a bipolar electrode (FHC Inc., Bowdoin, ME) placed into the middle molecular layer

(MML) approximately 150 um from the cells of interest, using a 0.1 msec 100 mA stimulating current. Stimulation efficacy was verified by recording from immediately adjacent mature cells (unlabeled) or by recording extracellular field potentials with an ACSF-filled glass pipette placed in the MML in the dendritic field adjacent to the immature neuron, while keeping the stimulation position and intensity constant. All recordings from mature cells or dendritic fields demonstrated clear responses to stimulation. Solutions and acute slice protocols are as reported previously (*Schnell et al., 2014*).

## Adult hippocampal neurospheres assay

Adult hippocampal neurospheres were derived from 6–8 microdissected dentate gyri of 4–6 weeks old WT mice. To isolate neural progenitors from the adult hippocampus, 6–8 adult dentate gyri were isolated under a microscope in dissection media (NaCl (137 mM); KCl (5.3 mM); HEPES (10 mM); D-glucose (33 mM); sucrose (44 mM); NaHPO$_4$·7H$_2$O (0.167 mM); KH$_2$PO$_4$ (0.220 mM); phenol red (0.067 mM); 1% pen/strep), then digested with papain for 30 min (10 ml Dissection medium; 2 mg cysteine; 150 µl calcium solution (100 µM); 100 µl EDTA solution (50 mM); 200 units papain). The digestion was stopped by re-suspension in Neurocult medium (Stem Cell Technologies, Canada), tissue was triturated to a single cell suspension and filtered by 70 µm cell strainer. Primary cells were seeded at a density of $2 \times 10^4$ viable cells/cm$^2$ in Neurocult medium supplemented with 20ng/ml hrEGF (Peprotech), 10 ng/ml hrbFGF (Peprotech, Rocky Hill, NJ) and 2 µg/ml heparin (Sigma). To induce hypoxia primary cells were cultured in a temperature- and humidity-controlled hypoxic chamber set at 2.5% O$_2$, 5% CO$_2$, and 94% N$_2$ (COY laboratory equipment, Grass Lake, MI). The apparatus contains a separate access chamber, as well as two pairs of work gloves, allowing manipulation of cultures in a continuously hypoxic environment. Alternatively, cells were treated every other day with hypoxia-mimetic agent DMOG (100 µM, Cayman Chemicals). After 7 days of culture the floating spheres were first attached to glass coverslips using Cell-Tak cell tissue adhesive (BD, Franklin Lakes, NJ) and then processed for further analysis.

Immunocytochemistry on neurospheres was carried out as previously described (*Varia et al., 1998*). Primary antibodies used included anti-Nestin (1:200, Millipore), anti-DCX (1:700, Millipore), anti-activated Caspase-3 (1:200, Abcam), anti-pimonidazole hydrochloride (1:200, Hypoxyprobe). Immunopositive cells and total DAPI-stained nuclei were counted to calculate the percentage of immunopositive cells. Confocal images of 10 randomly picked neurospheres from three independent experiments were counted for each marker and experimental condition. Neurospheres' diameter was calculated using Image J software.

## Statistical analysis

Statistical analysis was performed with Prism Software (GraphPad Software, La Jolla, CA). All data were obtained from n≥3 animals and expressed as mean ± standard deviation (SD). Experiments involving ≥ two groups were compared using unpaired two-tailed Students t-test or multiple-way analysis of variance (AVOVA) followed by post-hoc analysis with Sidak's multiple comparison test. Significance was defined as *p <0.05, **p <0.01, ***p <0.001.

# Acknowledgements

We would like to thank Dr. Ines Koerner for the generous use of the hypoxia chamber, Dr. Robert Hevner for the anti-Tbr2 antibody, Dr. Ben Emery and Antoinette Foster for their help with western blotting, and Lori Vaskalis for help with illustrations.

# Additional information

## Funding

| Funder | Grant reference number | Author |
|---|---|---|
| National Institutes of Health | NS R01 080979 | Gary L Westbrook |
| Ellison Medical Foundation | | Gary L Westbrook |
| National Institutes of Health | NS P30 061800 | Gary L Westbrook |
| AXA Research Fund | | C Chatzi |

| U.S. Department of Veterans Affairs | CDA-2 | E Schnell |

The funders had no role in study design, data collection and interpretation, or the decision to submit the work for publication.

## Author contributions

CC, ES, GLW, Conception and design, Acquisition of data, Analysis and interpretation of data, Drafting or revising the article

## Ethics

Animal experimentation: All procedures were performed according to the National Institutes of Health Guidelines for the Care and Use of Laboratory Animals and have been conducted with the approval of the Institutional Animal Care and Use Committee (#IP00000148) and the Insitutional Biosafety Committee (#04-06) at Oregon Health and Science University.

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
