## [Decision Letter]

[Editors’ note: this article was originally rejected after discussions between the reviewers, but the authors were invited to resubmit after an appeal against the decision.]

Thank you for choosing to send your work entitled "Differential oxygen tensions within the SGZ determine the early survival of newborn hippocampal granule cells" for consideration at *eLife*. Your full submission has been evaluated by Eve Marder (Senior Editor), a Reviewing Editor and two peer reviewers, and the decision was reached after discussions between the reviewers and Reviewing Editor. Based on our discussions and the individual reviews below, we regret to inform you that in its present form your work will not be considered further for publication in *eLife*.

You will note that one of the reviewers was quite enthusiastic about the manuscript and the other was much less so. Nonetheless, both of them raised some of the same or similar technical issues. These include the potential of hypoxic conditions introduced in the histological processing, the reduction of mRNA by a drug that supposedly inhibits at the protein level, and whether the data support "oxygen tension." Clearly, resolving these technical considerations is of paramount importance, and it isn't immediately obvious that this can be done in a very short time frame. If you believe that you can satisfactorily address these and the other issues raised in the reviews, the potential significance of the work would make us willing to consider a new submission of this work, which might be handled by the same or different reviewers.

*Reviewer #1:*The authors aim to show that migration from a hypoxic area in the SGZ to outer layers of the GCL is accountable for the death of young hippocampal neuroblasts after cell division. Although the idea of oxygen tension gradient in neurogenesis regulation is interesting and supported by previous data, this paper is far from providing proofs for this mechanism and the quality of images is not satisfactory. Many of the conclusions are not supported by the data.

My detailed remarks are listed here:

Figure 1: a major concern is about the proper tissue handling when using Hypoxyprobe. According to the Methods, animals were transcardially perfused with saline (subsection “Immunohistochemistry and Quantitation”). For how long were they perfused? It is of extreme importance because a brain not exposed to blood oxygen and not fixed yet with PFA will suffer from experimentally-induced hypoxia. It is crucial for the authors to show a similar Hypoxyprobe labeling pattern when the brains are immediately immersed in PFA.

Figure 1: the schematic should come at the end of the paper.

Figure 1 top right: how does this image correlate with the other images in the top row? Supposed to be an enlargement? It is not clear whether this cell is GFAP+ or not.

Figure 1: where are the 8-oxo-DG cells in the two images on the right?

The authors claim that DCX+ cells are located in more external layers of the GCL and away from the SGZ but in the images presented it appears that they are localized to the same place as Tbr2+ cells.

Overall, It would also be interesting to show how many of the NSCs are positive for Hypoxyprobe and whether it correlates with their activation status.

Figure 2: the regulation of prolyl hydroxylase on the levels of Hif1-α is at the protein level (by inducing its rapid degradation). Therefore what is the rationale for measuring Hif1-α mRNA levels? It should not be changed if prolyl hydroxylase is inhibited. Instead, the protein level should be measured.

Figure 2: it is not clear from the image which cells are pAKT+VE.

The legends repeat the conclusion of more pAKT cells twice.

Figure 3: the staining presented is not satisfactory. DCX is way too much overexposed and nestin staining probably represents non-specific blood vessel staining. It is not clear which cells are proliferating and no conclusion can be drawn.

Figure 3: are you sure these cells were labeled with BrdU (as indicated in y-axis)? If indeed please relate to the labeling protocol.

Figure 4: how were cell numbers normalized? Why not presenting cell numbers as # of positive cells per area or volume unit?

It is not clear to me why the numbers of Ki67 cells were not changed but the numbers of Tbr2 increased. These markers label the same population. Also quantification for Tbr2 should be added.

Figure 5: images in Figure 5 are not representative because there are more mCherry cells in the image but more GFP cells in the quantification. The images are not of good quality. There are many double-positive cells – how did you count them? Why are there so many red labeled cells in the hilus and outer GCL in *Bax* mutants treated with DMOG? It is not conclusive enough that the difference between *Bax* mutants treated with vehicle and *Bax* mutants treated with DMOG is due to increase in the fraction of mCherry cells. It rather can inhibit proliferation/survival of GFP cells.

Figure 5: why connect the groups with lines?

Figure 6: I do not understand why these experiments were conducted. It was shown before (by Gage group and others) the exact timing of synapse formation. The authors did not conduct the same electrophysiology experiments on DMOG-treated animals.

Figure 7: I also do not understand why the number of neurosphere is an indication for cell survival. Does DMOG or hypoxia increase the survival of the cells in the dish that initiate the sphere? Perhaps a better experiment would be to use a DMOG-treated animal for sphere production rather than to apply it to the dish.

*Reviewer #2:*This is a great study by Drs. Chatzi et al. The paper will have an important impact in the field of adult neurogenesis. The study is also highly significant for the field of hypoxia in general, including e.g. tumor biology. The study is particularly relevant for understanding the cognitive consequences of obstructive sleep apnea. The specificity with which the different stages of neurogenesis are affected by HIF/hypoxia is impressive.

My great enthusiasm for this paper is somewhat dampened by several aspects that need to be addressed before considering this paper for publication in *eLife*. Most of these comments relate to the hypoxic condition, which was primarily deduced from the use of molecular markers. One example is the title, which states "oxygen tensions within the SGZ[…]" yet, oxygen tensions were not really measured, but are suggested by the use of a molecular marker. I don't doubt the study, but the statement "Oxygen tensions" somehow implies that those oxygen tensions were measured. It is for example conceivable that these cells are just functioning anaerobically, i.e. they are metabolically hypoxic, but the tension may be normal? Changing the title would solve this issue. But, it is a recurrent issue, and the authors have to double check their statements: In the second section of Results, the authors state e.g. "Mimicking hypoxia". But, really what they (DMOG) do is stabilize Hif1-α. This, however, is only one aspect of hypoxia – hypoxia will also affect other regulators such as HIF2. Indeed, the balance between the regulation of oxidants and anti-oxidants is one of the aspects that would differentiate chronic versus intermittent hypoxia. Stabilizing HIF1a is not addressing this important aspect of hypoxia or the regulation of the Redox state, just to name one example. In short hypoxia will cause many additional biochemical changes throughout the cellular and local niche environments, which are not accounted for by Hif1-α stabilization. Again, this is easily fixed by changing the heading, and other statements that imply that the authors "mimicked hypoxia" – e.g. at the end of subsection “DMOG increases survival, but not proliferation or differentiation”.

In addition to this general comment, I have also a few specific comments.

1) Previous studies (e.g.,Hodge et al., 2008; Roybon et al., 2009) suggest that type 3 neurons can potentially be Tbr2+/DCX+. Therefore, it is recommended that the analysis be revised to pool type 2b with type 3 cells OR use Sox 11+/Prox1+ staining should the authors wish to resolve type 3 versus type 2.

2) Figure 3: it is unclear as to how the KI67+ cell counts were normalized. The text describes KI67+ cells/field. This should be clarified since the area of SGZ will vary between sections.

3) Given the ubiquity of HIF and the systemic application of DMOG, it would be useful for the reader to know whether stabilizing Hif1-α had any effects on macroscopic structures/volumes of the DG.

4) The statement "Double inmmunohistochemistry with an antibody that specifically recognizes phosphorylated Akt together with anti-DCX" is unclear.

5) Was the electrophysiology conducted in untreated animals? Please clarify. I also suggest that Figure 6 be placed in separate figures.

[Editors’ note: what now follows is the decision letter after the authors submitted for further consideration.]

Thank you for resubmitting your work entitled "Localized hypoxia within the SGZ determines the early survival of newborn hippocampal granule cells" for further consideration at *eLife*. Your revised article has been favorably evaluated by Eve Marder (Senior Editor), a Reviewing Editor, and three reviewers.

All reviewers are generally supportive of the potential value of your work and its suitability for publication in *eLife*. However, the consensus opinion of the reviewers is an additional set of control experiments is required before your paper can be published. These studies are meant to: (a) provide additional clarifying data on the exact cell types affected in the transition from more hypoxic to less hypoxic regions of the hippocampus; (b) improve confidence in the specificity of effects of systemically administered DMOG; (c) provide additional data supporting your conclusions about the lack of influence of synaptic activity on progenitors.

Specifically, here are the experiments suggested by the reviewers:

*Reviewer 2:* 1) For Figure 4, immunohistochemical analysis should be provided for Nestin/BrdU and Tbr2/Dcx at all time-points in which BrdU was measured. This is in line with analysis shown in Figure 3 and should be feasible for the authors.

2) For Figure 4, analysis done at 28 d.p.i specifically should include a BrdU/NeuN analysis since the major conclusion addresses survival.

*Reviewer 3:*

1) The authors’ evidence against neural progenitors receiving any synaptic inputs is weak compared to the evidence published arguing for a role of synaptic activity in modulating survival of NPCs (see Song et al., Nature Neuroscience, 2013). For instance, the authors can record from 3 day old NPCs while stimulating hilar INs or mossy cells.

2) Retroviral expression of Hif1-α in NPCs to address non-specific effects of DMOG.

3) Examining c-Fos in mossy cells, hilar INs at different time points during and immediately post DMOG treatment.

4) Western blots for Hif1-α in DG of DMOG treated brains.

We hope that you agree that this limited set of experiments are reasonable and will add to the quality of your paper.

---

## [Author Response]

[Editors’ note: the author responses to the first round of peer review follow.]

We were gratified that reviewer 2 called the paper “great”, but concerned by the views of reviewer 1. The rejection was based on “technical issues” that you mentioned in your letter. Specifically, the editors’ letter mentioned 3 technical issues:

1. “…The potential of hypoxic conditions introduced in the histological processing”. This is not a relevant concern based on the chemistry of the probe as we elaborate below. The methods of tissue fixation were entirely standard and in fact we had also done the less-ideal method proposed by the reviewer (immersion fixation of the entire brain) and the results were the same. We explain this further in the proposed point-by-point response to the reviewer as below.

2. “…The reduction of mRNA by a drug that supposedly inhibits at the protein level…”. The reviewer was not aware the Hif1a transcription can be autoregulated because of the presence of hypoxic response elements (HREs) in its promoter. Thus our analysis and interpretation are correct. In any case, this was a minor issue that is not critical to the overall thrust of the paper.

3. “…Whether the data support ‘oxygen tension’…” as what we measured. This point was raised by reviewer 2, and we agree that we have adjust the wording in the title, headers and text. However we used a hypoxic biosensor that we calibrated in vitro thus we think we are able to assess that regions of the subgranular zone have an oxygen tension less than 1.5%. It is true we did not use an oxygen electrode, but rather a readout of a chemical reaction.

Reviewer #1:

The authors aim to show that migration from a hypoxic area in the SGZ to outer layers of the GCL is accountable for the death of young hippocampal neuroblasts after cell division. Although the idea of oxygen tension gradient in neurogenesis regulation is interesting and supported by previous data [...].

This comment makes it sound like our paper investigated a well-understood mechanism. We disagree. Mazudmar et al. (2010) showed that oxygen regulates neural stem cells through Wnt-β catenin signaling in vitroand that deletion of hypoxia-inducible factor-1 α subunit (HIF-1α) in vivodecreased adult hippocampal neurogenesis. However the mechanisms and the developmental stages at which oxygen gradients affect adult neurogenesis remain unknown. The point of our paper was to show in vivothat heterogeneous oxygen gradients influence the survival of an early stage of neurogenesis, well before the better-studied activity-dependent survival that occurs later. These papers were cited and the topic discussed in the Introduction of the manuscript.

*[…] this paper is far from providing proofs for this mechanism and the quality of images is not satisfactory. Many of the conclusions are not supported by the data.*

We are able to easily address all specific comments as below.

*My detailed remarks are listed here: Figure 1: a major concern is about the proper tissue handling when using Hypoxyprobe. According to the Methods, animals were transcardially perfused with saline (subsection “Immunohistochemistry and Quantitation”). For how long were they perfused? It is of extreme importance because a brain not exposed to blood oxygen and not fixed yet with PFA will suffer from experimentally-induced hypoxia. It is crucial for the authors to show a similar Hypoxyprobe labeling pattern when the brains are immediately immersed in PFA.*

In brief, this is not a significant concern. Hydroxyprobe (pimonidazole) was delivered as a single injection and the animal was sacrificed at 60 minutes. Given that the half-life of Hydroxyprobe is 22 minutes in mice, >99% of free drug has been cleared by the time of sacrifice. Furthermore, pimonidazole becomes reductively activated and its active intermediate binds to thiol groups in nucleic acids and proteins in hypoxic cells. Thus bound drug is not in rapid equilibrium with free drug, i.e. the pimonidazole imaged in our experiments was already bound at time of perfusion.

In terms of perfusion, a saline rinse (4 ml, <30 sec) is standard protocol to remove blood and thus increase access of PFA to the brain, and transcardial perfusion is regarded as the most effective way to achieve rapid and complete fixation within 1 minute. Thus the saline rinse should reduce the chance of generalized hypoxia. Finally, if there was any free pimonidazole and if there was generalized hypoxia, we should have seen generalized staining as well as signs of generalized hypoxia – we did not. In addition, rapid immersion of the brain in PFA produced similar results (data not shown). Details of the kinetics of Hydroxyprobe have been added to Methods and references added.

*Figure 1: the schematic should come at the end of the paper.*

The schematic was introduced as the framework for the experiments, not a cartoon of the conclusions. We think this was clear, however we have modified the figure legend and text to make this explicit.

*Figure 1 top right: how does this image correlate with the other images in the top row? Supposed to be an enlargement? It is not clear whether this cell is GFAP+ or not.*

Yes, the rightmost panels were enlargements. For example, in the top row the rightmost panel shows a hypoxic cell labeled with pimonidazole (green) colocalized with the stem cell marker GFAP (yellow). The figure has been modified for clarity as has the figure legend. The point of this figure was clearly quantified in Figure 1.

*Figure 1: where are the 8-oxo-DG cells in the two images on the right? The authors claim that DCX+ cells are located in more external layers of the GCL and away from the SGZ but in the images presented it appears that they are localized to the same place as Tbr2+ cells.*

We did not claim that DCX+ cells are located in “more external layers of the GCL,” as that is not how this tissue is organized. DCX expressing immature newborns neurons in theDG are localized in the SGZ zone and inner GCL layer as clearly depicted in our images. Newborns cells migrate in the more external layer of the GCL much later during their development ( >2 weeks post-mitotic). We included the schematic in Figure 1 specifically to avoid such misunderstandings. Thus, as stated in the original text and obvious from the image, the 8-oxo-DG cells are located within the SGZ in the same layer with Tb2+and Tbr2+/DCX+ cells.

*Overall, It would also be interesting to show how many of the NSCs are positive for Hypoxyprobe and whether it correlates with their activation status.*

In our experiments, the vast majority of pimonidazole+ cells were neural stem cells (GFAP+), a cell type that can exist for long periods. Thus our labeling provides a single snapshot with a window of <60 minutes (the duration of free pimonidazole in the tissue). It is not practical to count GFAP+ cells because the labeling even in confocal sections is dense, but as clearly shown in the images only a small fraction of GFAP+ cells were pimonidazole-positive. For the more short-lived intermediate progenitors (Tbr2+), 14.6 ± 1.1% of Tbr2+ cells (Type 2a cells, 100 Tbr2+ cells from 3 animals) were double labeled with pimonidazole. We did not see colabeling of pimonidzole with Ki67 (as was mentioned in the Discussion). We have further elaborated on this point in the Discussion.

*Figure 2: the regulation of prolyl hydroxylase on the levels of Hif1-α is at the protein level (by inducing its rapid degradation). Therefore what is the rationale for measuring Hif1-α mRNA levels? It should not be changed if prolyl hydroxylase is inhibited. Instead, the protein level should be measured.*

This is not correct. Several hypoxia response elements (HREs) have been identified in the *HIF1A* promoter and implicated in an autoregulatory loop whereby HIF itself upregulates HIF-1α transcription. Thus stabilization of Hif1-α by DMOG can in turn induce an increase in Hif1-α mRNA levels. A specific anti-Hif1-α antibody is also not commercially available. This point has been added to the Results and references added.

*Figure 2: it is not clear from the image which cells are pAKT+VE. The legends repeat the conclusion of more pAKT cells twice.*

Arrows in the original figure clearly identified the pAKT cells, but the figure has now been enlarged to make this point even clearer. The figure legend was not repetitive, it merely identified the raw data panel and the quantification in the histogram. However, we have now consolidated this wording.

*Figure 3: the staining presented is not satisfactory. DCX is way too much overexposed and nestin staining probably represents non-specific blood vessel staining. It is not clear which cells are proliferating and no conclusion can be drawn.*

We completely disagree. Nestin clearly labeled the linear radial glia fibers as expected, and thus is definitely not non-specific. It is true that Nestin labeled blood vessels in the hilus, but their distinctive morphology (horizontal vs vertical in this image) made them clearly separable. Because we used a mouse antibody, we expect blood vessel staining but this has no impact on the quantification. To avoid confusion, Figure 3 has been modified to show only Ki67 immunolabeling at low power (panels A, B), and triple staining with Ki67/Nestin/DCX at higher magnification (panels C, D).

*Figure 3: are you sure these cells were labeled with BrdU (as indicated in y-axis)? If indeed please relate to the labeling protocol.*

Yes. BrdU methodology was in the original Methods section. BrdU methods have been added to the figure legend and we have clarified the text in the subsection “DMOG increases survival, but not proliferation or differentiation”.

*Figure 4: how were cell numbers normalized? Why not presenting cell numbers as # of positive cells per area or volume unit?*

Cell counts were normalized to the vehicle-treated controls. We have now converted these numbers and plotted the figure as positive cells per tissue volume (cells/mm^3^), which has no effect on the conclusions.

*It is not clear to me why the numbers of Ki67 cells were not changed but the numbers of Tbr2 increased. These markers label the same population. Also quantification for Tbr2 should be added.*

We did not detect (or claim) a significant effect on the total density of the Tbr2+ cells; thus there is no discrepancy. We have added the data on Tbr2 (Vehicle: 5776 ± 681 cells/mm^3^, DMOG: 7331 ± 1381 cells/mm^3^, p=0.07, n= 5 animals) in the subsection “DMOG increases survival, but not proliferation or differentiation”.

*Figure 5: images in Figure 5 are not representative because there are more mCherry cells in the image but more GFP cells in the quantification. The images are not of good quality. There are many double-positive cells – how did you count them? Why are there so many red labeled cells in the hilus and outer GCL in* Bax *mutants treated with DMOG?*

The images were representative, but the reviewer has interpreted the relatively high background of the red channel in low magnifications as mCherry+ cells. There are no mCherry+ cells in the hilus and outer GCL. This is not a question of the quality of the image but due to the fact that the animals were analyzed one week post stereotaxic injections as necessary for the experiment, and mCherry in vivomatures much slower than GFP. Thus the signal/noise ratio is not as good in the red channel than in the green channel at low mag. We used high magnification images for quantification where mCherry+ cells are easily resolved as was apparent in the high mag images in the original figure. All cells including double labeled cells were counted. We have worked to improve the image quality at low magnification given the technical constraints as discussed here.

*It is not conclusive enough that the difference between* Bax *mutants treated with vehicle and* Bax *mutants treated with DMOG is due to increase in the fraction of mCherry cells. It rather can inhibit proliferation/survival of GFP cells.*

The two retrovirus strategy was used simply to allow quantification and thus these are the same population of newborn neurons. If, as the reviewer implies, DMOG inhibited proliferation or survival, GFP+ and mCherry+ cells would be equally affected and the ratio of GFP/mCherry cells unchanged. Thus we do not see this as a logical explanation of the data. We have amplified this point in the subsection “DMOG increases survival, but not proliferation or differentiation”.

*Figure 5: why connect the groups with lines?*

The panel has been replotted as a histogram.

*Figure 6 do not understand why these experiments were conducted. It was shown before (by Gage group and others) the exact timing of synapse formation. The authors did not conduct the same electrophysiology experiments on DMOG-treated animals.*

There is conflicting evidence in the literature concerning the timing of first synaptic activity and its influence on cell survival. Given that even small amounts of neural activity can have an impact, we think high quality data in this regard is critical and it is key to our conclusion that this early period of cell survival is independent of neural activity. It could be a supplemental figure, but given *eLife’s* format we think it is better included as a regular figure. We see no rationale for repeating the electrophysiology in DMOG-treated animals, given that DMOG did not affect the surrogate activity markers pCREB or c-Fos.

*Figure 7 also do not understand why the number of neurosphere is an indication for cell survival. Does DMOG or hypoxia increase the survival of the cells in the dish that initiate the sphere?*

The number of neurospheres is a standard measure in the stem cell field to assess survival. Because the seeding density of dissociated cells for neurosphere formation was the same for all conditions, the total number of neurospheres reflects the survival of the progenitors.

*Perhaps a better experiment would be to use a DMOG-treated animal for sphere production rather than to apply it to the dish.*

We don’t see this as an interpretable experiment. The reason for using neurospheres is that we can control the oxygen levels and the concentration of DMOG. The in vivotissue oxygen concentration is much lower than the oxygen concentration used for routine cell cultures (room air, 21%). The ability to compare the effects of real hypoxic conditions with DMOG treatment on progenitors in vitroprovides a valuable calibration tool with clean readouts.

Reviewer #2:

*My great enthusiasm for this paper is somewhat dampened by several aspects that need to be addressed before considering this paper for publication in* eLife*. Most of these comments relate to the hypoxic condition, which was primarily deduced from the use of molecular markers. One example is the title, which states "oxygen tensions within the SGZ[…]" yet, oxygen tensions were not really measured, but are suggested by the use of a molecular marker. I don't doubt the study, but the statement "Oxygen tensions" somehow implies that those oxygen tensions were measured. It is for example conceivable that these cells are just functioning anaerobically, i.e. they are metabolically hypoxic, but the tension may be normal? Changing the title would solve this issue. But, it is a recurrent issue, and the authors have to double check their statements: In the second section of Results, the authors state e.g. "Mimicking hypoxia". But, really what they (DMOG) do is stabilize Hif1-α. This, however, is only one aspect of hypoxia – hypoxia will also affect other regulators such as HIF2. Indeed, the balance between the regulation of oxidants and anti-oxidants is one of the aspects that would differentiate chronic versus intermittent hypoxia. Stabilizing HIF1a is not addressing this important aspect of hypoxia or the regulation of the Redox state, just to name one example. In short hypoxia will cause many additional biochemical changes throughout the cellular and local niche environments, which are not accounted for by Hif1-α stabilization. Again, this is easily fixed by changing the heading, and other statements that imply that the authors "mimicked hypoxia" – e.g. at the end of subsection “DMOG increases survival, but not proliferation or differentiation”*.

We have changed the title, wordings and section headers to address this issue. We agree with the reviewer that this does not change the conclusions of the manuscript.

*In addition to this general comment, I have also a few specific comments. 1) Previous studies (e.g., Hodge et al., 2008; Roybon et al., 2009) suggest that type 3 neurons can potentially be Tbr2+/DCX+. Therefore, it is recommended that the analysis be revised to pool type 2b with type 3 cells OR use Sox 11+/Prox1+ staining should the authors wish to resolve type 3 versus type 2.*

We agree that our staining paradigm cannot differentiate between Type 2b and Type 3 cells. We have revised the analysis and data have been plotted as percentages of Tbr2+/DCX-, Tbr2+/DCX+ and Tbr2-/DCX+ cells and the text reworded in the subsection “DMOG increases survival, but not proliferation or differentiation”.

*2) Figure 3: it is unclear as to how the KI67+ cell counts were normalized. The text describes KI67+ cells/field. This should be clarified since the area of SGZ will vary between sections.*

Ki67 cells counts are now expressed as fold increases of Ki67cells/mm^3^ in DMOG vs. vehicle-treated animals. For normalization the density of Ki67+ cells in individual DMOG treated animals was normalized to the mean cell density of the animals in the control group (see Methods).

*3) Given the ubiquity of HIF and the systemic application of DMOG, it would be useful for the reader to know whether stabilizing Hif1-α had any effects on macroscopic structures/volumes of the DG.*

Administration of DMOG had no effect on the DG volume (Control: 0.61mm^3^ ± 0.02, n=3 animals, DMOG: 0.59 ± 0.04, n=animals, p=0.89). This point has been added to the text in the subsection “DMOG increases survival, but not proliferation or differentiation”.

*4) The statement "Double inmmunohistochemistry with an antibody that specifically recognizes phosphorylated Akt together with anti-DCX" is unclear.*

The statement has now been better explained in the subsection “Stabilizing Hypoxia Inducible Factor-1α” as follows: “Double immunohistochemistry with an antibody that specifically recognizes phosphorylated Akt together with an antibody against the neuroblast/immature neuron marker doublecortin (DCX)[…]”

*5) Was the electrophysiology conducted in untreated animals? Please clarify. I also suggest that Figure 6 be placed in separate figures.*

Yes, the electrophysiology was done in control animals as the question was whether there is any synaptic activity at this stage. We prefer to keep the activity-related panels together in one figure, but we have clarified that Figure 6 refers to the lack of electrical activity in 3-day-old cells in control animals and Figure 6 address the effect of DMOG on overall activity, which addresses whether non-cell autonomous neural activity could explain the effect of DMOG. This point has been further clarified in the subsection “Role of neuronal activity“.

[Editors’ note: the author responses to the re-review follow.]

Reviewer 2:

*1) For Figure 4, immunohistochemical analysis should be provided for Nestin/BrdU and Tbr2/Dcx at all time-points in which BrdU was measured. This is in line with analysis shown in Figure 3 and should be feasible for the authors.*

The reviewer wanted us to document the phenotype of BrdU+ cells at each of the time points used in the survival experiments. This is now included in Figure 3. We used triple immunolabeling for Tbr2/DCX/BrdU. The data has been plotted as percentages of Tbr2+/DCX-, Tbr2+/DCX+, Tbr2-/DCX+ and unlabeled cells (i.e. mature cells at 21 and 28 dpi that no longer label with DCX or Tbr2, and rare stem cells at dpi 3). Nestin and Tbr2 antibodies were raised in the same animal and thus could not be used together, but Tbr2/DCX/BrdU address the same issue. These additional experiments do not alter any of the conclusions. (Figure 3, Figure 3 legend and the text in the subsection “DMOG increases survival, but not proliferation or differentiation” have been revised.)

*2) For Figure 4, analysis done at 28 d.p.i specifically should include a BrdU/NeuN analysis since the major conclusion addresses survival.*

We have now included BrdU/NeuN analysis at 28 d.p.i. for animals exposed to vehicle or DMOG for the first 3 and 7 days post-mitosis – the two time-windows during which DMOG promoted survival of newborn cells. At 28 d.p.i. >95% of BrdU+ cells were also NeuN+ follow 3 or 7 day treatment with vehicle or DMOG. These experiments provide further support that surviving cells fully differentiate into mature neurons by 28 days post mitosis. For the convenience of the reviewer, we have included Figure 8 with this same data, which is now included in the text in the subsection “DMOG increases survival, but not proliferation or differentiation”.

Author response image 1.The percentage of BrdU+ cells in the GCL that was positive for the mature neuronal marker NeuN at 28 d.p.i. was similar between vehicle or DMOG treated animals during the two time-windows (0-3 and 0-7 days) of treatment that promoted survival of newborn neurons.(0-3 days, Vehicle: 96 ± 4, DMOG: 95 ± 2 n=3; 0-7 days, Vehicle: 98 ± 3, DMOG: 96 ± 2, n=3).**DOI:**
http://dx.doi.org/10.7554/eLife.08722.010

Reviewer 3:

*1) The authors’ evidence against neural progenitors receiving any synaptic inputs is weak compared to the evidence published arguing for a role of synaptic activity in modulating survival of NPCs (see Song et al., Nature Neuroscience, 2013). For instance, the authors can record from 3 day old NPCs while stimulating hilar INs or mossy cells.*

We strongly disagree that our evidence was “weak”. We also think the reviewer overstates the prior evidence, which was at 4 days post-mitosis in which they state that “14%” of their cells had some activity at high stimulation intensity, but the example shown in that paper does not fit with the expected kinetics of a GABAergic response. Thus we don't know how to interpret that data.

In any case, we did not find any evidence for synaptic innervation of immature adult-born granule cells at 3 days post-mitosis, based on assays of spontaneous events and the lack of response to high intensity 8 Hz stimulation synaptic responses. Reviewer 3 suggests that stimulation within the middle molecular layer (MML) could have missed possible synaptic connections arising from hilar cells (interneurons or mossy cells). However, hilar basket cells and hilar mossy cells are driven by excitatory feed-forward inputs from the perforant path (which we stimulated directly) as well as inputs from dentate granule cell mossy fiber axons (see Ribak, CE and Shapiro, LA (2007) Ultrastucture and connectivity of cell types in the adult dentate gyrus. in Scharfman, H.E (ed) The Dentate Gyrus, 157-160). To prove explicitly that our stimulation activates hilar cells, we stimulated the MML using the same stimulating electrode position, intensity, and train duration/frequency as shown in the manuscript and assayed hilar neurons with cell-attached and current-clamp recordings (n=4 cells). In every cell, we elicited depolarization, and in 3 of 4 cells, the cells showed repetitive spiking (Figure 9). Thus hilar cells were activated in the protocol shown in the manuscript, yet no neural activity in the 3 day-old cells was observed. We think this evidence is unequivocal that the cells at 3 d.p.i show no neural activity, thus justifying the conclusions in our manuscript. This point is now included in the subsection “Role of neuronal activity”.

Author response image 2.Hilar neuron activation by MML 8 Hz stimulation.A bipolar stimulating electrode was placed in the middle molecular layer of the dentate gyrus of a mouse hippocampal slice, and a whole-cell current clamp recording was made from a hilar neuron. In this example, the hilar neuron spiked reliably in response to each stimulation episode (dot above trace). Scale bars, 20mV, 0.5 sec.**DOI:**
http://dx.doi.org/10.7554/eLife.08722.011

2) Retroviral expression of Hif1-α in NPCs to address non-specific effects of DMOG.

This is not a practical experiment, as expression of a construct requires several days post-retroviral injection, thus assay at 3 days would not be interpretable. Even if one was able to overexpress Hif1-α, it is rapidly degraded in normoxia, which is exactly the reason that DMOG (which inhibits the hydrolase that degrades Hif1-α) is useful. With regard to “non-specific” effects of DMOG, this reagent has been well studied in other systems and has proven to be highly specific.

*3) Examining c-Fos in mossy cells, hilar INs at different time points during and immediately post DMOG treatment.*

The manuscript showed that DMOG did not affect neuronal activity in the dentate as measured by c-Fos immunohistochemistry at 3 d.p.i. The reviewer wanted us to also show similar data for hilar neurons. Thus we performed double immunohistochemistry with c-Fosand the GABAergic marker glutamic acid decarboxylase-67 (Gad-67). This paradigm allowed us to differentiate between activated hilar mossy cells (Fos+ only) and active hilar interneurons (Fos+/GAD67+). In general were very few activated (Fos+) hilar mossy cells or interneurons in the vehicle-treated animals (Vehicle: Fos only+: 144 ± 55 cells/mm^3^, n=4, Fos+/GAD67+: 14 ± 16 cells/mm^3^, n=4). DMOG treatment did not affect the density of Fos+ cells in the hilus (DMOG,Fos only+: 116 ± 14 cells/mm^3^, n=4, Fos+/GAD67+ve: 13 ± 14 cells/mm^3^, n=4, p=0.4). These data further support our findings that DMOG’s effect on survival is activity-independent. These data are now included in the subsection “Role of neuronal activity”.

*4) Western blots for Hif1-α in DG of DMOG treated brains.* We have added Western blots of Hif1-α in whole dentate gyrus to Figure 2 (new panel B). The data are consistent with the gene expression level in the original manuscript and show that Hif1-α protein is higher in DMOG treated animals. This was also predicted from the data in the original manuscript showing changes in mRNA of downstream targets of HIF1a, an effect that requires changes in Hif1-α protein. Text describing these data has been added to the subsection “Stabilizing Hypoxia Inducible Factor-1α”.